# Analysis of the Spatiotemporal Pattern and Mechanism of Land Use Mixture: Evidence from China's County Data

**Yanting Zheng** [1,2], **Sai Zhao** [1], **Jinyuan Huang** [1,2] and **Aifeng Lv** [3,4,*]

1 School of Economics and Resource Management, Beijing Normal University, Beijing 100875, China; zhengyt@bnu.edu.cn (Y.Z.); 201931410007@mail.bnu.edu.cn (S.Z.); 201921410003@mail.bnu.edu.cn (J.H.)
2 Beijing Key Lab of Study on SCI-TECH Strategy for Urban Green Development, Beijing 100875, China
3 Institute of Geographic Sciences and National Resources Research, Chinese Academy of Sciences, Beijing 100101, China
4 University of Chinese Academy of Sciences, Beijing 100049, China
* Correspondence: lvaf@igsnrr.ac.cn

**Abstract:** The mixture of agricultural and non-agricultural land-use represents a new pattern of urbanization in the Global South. This mixture has hindered the improvement of land-use productivity and makes it difficult to achieve the centralized disposal of pollutants, which has resulted in the waste of land resources and serious environmental problems. Although many studies have investigated land-use mixture, most of them remain descriptive and lack quantitative examination and an in-depth mechanism analysis. Using raster land-use data, this paper examines the spatiotemporal pattern of the land-use mixture in China between 1990 and 2015 by calculating join counts values supplemented by landscape metrics, and attempts to explain the regional variations in land-use mixtures in recent years. The results show that, between 2000 and 2010, land-use was more mixed in fast-growing regions such as Zhejiang, Fujian, Chongqing, Guangdong, and some major metropolises and mining cities, and that, between 2010 and 2015, land-use was more mixed in Central China. Additionally, the results of econometric models reveal that mixed land-use can be alleviated in regions with strict land planning and management, such as urban agglomerations in the Yangtze River Delta and the Pearl River Delta, as well as in areas with high levels of urbanization. Furthermore, the results of a spatial heterogeneity analysis show that strict land management has played an important role in reducing the land-use mixture in Eastern China; however, it has not played a significant role in Central China. The findings of this study suggest that land-use should be appropriately planned and managed to ensure sustainable development.

**Keywords:** land-use mixture; non-agricultural land; planning and management; join counts method; landscape metrics

## 1. Introduction

China's unprecedented industrialization and urbanization since the late 1980s have resulted in a large amount of non-agricultural land conversion [1–4]. In 1980–2010, the area of built-up land in China increased by 5.52 million hectares [1], with the value in 2012 being four times that in 1985 [5]. This large-scale and uncontrolled land-use transformation has brought unprecedented challenges to China's urban planning and management. The disorderly expansion of non-agricultural land combined with poor management has led to agricultural land being invaded and occupied in a fragmented way, and agricultural and non-agricultural land are often spatially mixed [6–12]. The land-use mixture can be defined as the degree to which large areas of two different land-uses (e.g., agricultural and non-agricultural land-uses) coexist in the same area, often in a spatially piecemeal manner [13,14]. An intensive land-use mixture makes it impossible to realize economies of scale in agriculture and industry and severely hinders the improvement of agricultural

and industrial productivity [9]. Additionally, the mixture of agricultural land and non-agricultural land complicates the sharing of public infrastructure and makes it impossible to realize the centralized disposal and the recycling of pollutants such as PM2.5 and sewage, which brings serious environmental pollution problems [8,10,15].

Research on the land-use mixture can be traced back to McGee's (1991) [16] study of desakota (desa means village in English and kota means town in English) in Asian countries. This study found that industrialization in Asia does not necessarily occur in cities, as it did in the Western world in the early 19th century, but can occur at a large scale in rural areas. Compared with cities, which have higher land prices, rural areas have more available cheap land and a lower level of environmental supervision [17]. These conditions are often favored by enterprises, especially foreign ones, who do not care about the urban condition of the industrial location but rather prefer large plots and single-story factories. As a result, many enterprises have appeared in rural areas in China. The fragmented governance of rural areas causes the locations of firms to become more scattered and the juxtaposition of agricultural and non-agriculture lands to become widespread. The urban–rural hybrid configuration in McGee's conceptual model occurs widely in various contexts in the Global South. The model is even believed responsible for disorienting the worlding of urban–rural dualisms and city-centric urban studies [18].

Various studies have examined landscape changes arising from the economic-driven fragmentation of landscapes in urban peripheries [11,19,20]. Some quantitative studies used landscape metrics to measure the degree of landscape fragmentation [21]. These investigations often focused on one land-use type and their landscape indicators were often calculated based on mathematical formulas [5,6,22,23]. However, the fragmentation of a single land-use type may not exactly represent the mixture of two land-use types, or the ambiguous spaces juxtaposing urban and rural landscapes, and the mathematical formulas may not be sufficient to clearly describe the meaning of the land-use mixture. Some studies have tried to analyze the fragmentation of all land-use types [24]; however, these do not reflect the extent to which agricultural land is occupied by non-agricultural land. In addition to landscape metrics, some scholars have used the join counts method to carry out targeted measurement of desakota according to its characteristics. Murakami et al. (2005) [8] used the join counts method to measure the land-use mixture in Asian megacities. Furthermore, Zheng (2009) [12] used the same method to study the land-use mixture in the urban agglomerations in China, confirming that the method can accurately describe the land-use mixture. However, these studies mainly focused on the measurement of the emergence of desakota, lacking a large national-scale and quantitative inquiry of the mechanisms behind it. In this study, we maintain that the join counts method based on the number grid principle, in which the joins between contiguous different grid cells are counted, conforms to the definition of the land-use mixture and also conforms to the description of the land-use juxtaposition of McGee (1991) [16]. Therefore, this study used the join counts method with several general landscape metrics as supplements to reflect the land-use mixture and determine where this phenomenon occurs in China, how it has changed over time, and what factors have led to it.

In this paper, we empirically investigate the above questions. We examined which Chinese counties experienced significant changes in their land-use mixture from 1990 to 2015 and attempted to compare the impacts of urban and rural development on the land-use mixture and, additionally, understand the role of institutional factors on these changes. For example, is urban or rural expansion more likely to lead to the land-use mixture, and how do the land management intensity, fiscal decentralization, and other factors affect the land-use mixture?

This analysis adds to an extensive body of research on non-agricultural land juxtaposition, including studies by Gottmann (1961) [25], McGee (1991) [16], and Zhou (1991) [26]. Additionally, we aimed to expand upon recent studies on the change of China's land-use and the mechanisms behind this change. Many researchers have studied changes in the urban land in China, noting that the rate of expansion of urban land has accelerated across

the country in recent years [27–29]. These studies found evidence that population growth, economic growth, and national policies have promoted the expansion of urban land.

With the unprecedented urbanization, China's large, medium, and small cities have expanded rapidly [30]. According to Fang et al. (2016) [5], from 1990–2005, the increase in urban non-agricultural land was dominated by urban in-filling and edge expansion. Therefore, it seems that urban expansion is not the main cause of the growth of the land-use mixture in China. Meanwhile, due to fragmented governance in rural areas, a large number of rural settlements and township enterprises are widely spread throughout the countryside with a dispersed distribution; therefore, rural expansion may lead to the increase in the land-use mixture [31]. However, there is still a lack of convincing quantitative research on the impact of urban and rural expansion on the land-use mixture.

Additionally, this article also considers the impact of loose planning regulations on the land-use mixture. This question was closely investigated by He et al. (2016) [32], who found that fiscal decentralization, political centralization, and poor land management have led to land urbanization in China. Similar arguments were made by Tian (2015) [7], who qualitatively analyzed land-use dynamics and revealed that the loose planning regulation over collective land is a key cause of fragmented and scattered industrial land-use. Collectively owned rural land, which is characterized by unfixed ownership (i.e., often managed by a three-tiered governance system consisting of the townships, administrative villages, and natural villages [33]), has experienced a government regulation vacuum [34]. Thus, in China, the lack of systematic management of collective lands has led to a booming informal land market and uncontrolled land development [7]. Although state agencies have usually acted as enforcers of land-use control, state rules usually cannot be effectively implemented due to conflicting interests [35]. Moreover, these rules have often been contested, circumvented, and manipulated by land developers and users, and state agencies and managers. This has led to uncontrolled land development in China, and consequently, a large number of illegal land development and land violation cases have been reported [36–39]. However, most previous studies primarily focused on the qualitative analysis of the urban landscape, and there has been a lack of quantitative analyses.

Many researchers have studied land-use at the provincial- or prefecture-level [32,40,41]. However, as Deng et al. (2008) [29] argued, land-use planning is mainly carried out at the county level. The county is the main economic and social unit in China and has been generally accepted as a stable economic unit since the Qin dynasty. Therefore, we consider that the county is an appropriate unit for studying land-use. In this paper, we examine the land-use mixture and its mechanism at the county level in China.

Additionally, due to the different levels of economic development in various regions of China, there are also large regional differences in the intensity of land management. Therefore, in this study, we used regional comparisons to emphasize the differences in both the spatiotemporal variations of land-use mixtures and their possible mechanisms in Eastern, Central, and Western China.

The Section 2 describes the materials and methods used in this study; the Section 3 illustrates the regional differences and spatial evolution of land-use mixtures and presents our empirical results; the Section 4 discusses the results for land-use mixtures; and the Section 5 summarizes our conclusions.

## 2. Materials and Methods

### 2.1. Study Area

This study considered 2861 districts and counties in China as the research units to examine the situation of the land-use mixture from 1990 to 2015. Due to limited data availability, none of the districts and counties in Hong Kong, Macau, and Taiwan were included. Since the 1980s, China has experienced unprecedented economic growth. In 2015, China's GDP reached 68.55 trillion yuan, its total population reached 13.75 billion, and its urbanization rate reached 56.1%. At the same time, rapid industrialization has significantly changed China's land-use structure. In 2015, China's construction land area

was 17.70 million hectares, representing an increase of 57.8% compared to 1980. Moreover, in 2015, the areas of forest land, grassland, and unused land in China were lower by 8.52 million hectares, 5.32 million hectares, and 1.15 million hectares compared to 1980, respectively [42].

### 2.2. Data and Variables

In this paper, we used raster land-use data sets with a resolution of 100 m for 1990, 2000, 2010, and 2015. The raster data were obtained from the China Land Use and Land Cover Change database (CNLUCC) at a 1:100,000 scale [43]. The aim of this article is to use the indicator of the land-use mixture to reflect the extent to which agricultural land is occupied by non-agricultural land in China. We did not pay attention to the mixed situation of all 25 types of land-use, but only focused on the mixture of agricultural land and construction land. Using the ArcGIS software, we classified the 25 types of land-use into two types: built-up areas (e.g., residential areas, industrial areas, roads) and agricultural areas (e.g., cropland, woodland, grassland). Water bodies (e.g., sea, rivers, lakes) are neither built-up areas nor agricultural areas, so they were not included in subsequent calculations. On this basis, the join counts method and some general landscape metrics were used to measure the land-use mixture. The specific methods are described in the following.

### 2.2.1. Join Counts Method

As illustrated in Figure 1, based on the raster land-use map with three land-use types (left), the number of joins between built-up areas and agricultural areas was calculated as the join counts value, and such values were assigned to each built-up area pixel (right). The assignment rules for built-up area cells were as follows: (1) take the built-up area cell as the central cell; (2) determine the number of agricultural area cells in the eight surrounding cells; (3) the assigned join counts value of the central cell is the number of agricultural area cells (i.e., the number of joins between the built-up area cells and the agricultural area cells). According to this procedure, the assigned join counts values of cells were between 0 and 8. For example, in the right-hand panel of Figure 1, there is a built-up cell with a value of 4. When this cell is used as the center cell, four of the eight surrounding cells are agricultural areas, one is a water body, and the other three are built-up areas. According to the assignment rules, the join counts value of the center cell is the number of agricultural land cells that surround it, which is 4. The join counts value indicates the frequency of the contiguity between built-up and agricultural areas, that is, the degree of the land-use mixture [8]. A higher join counts value indicates a higher degree of the land-use mixture around that pixel.

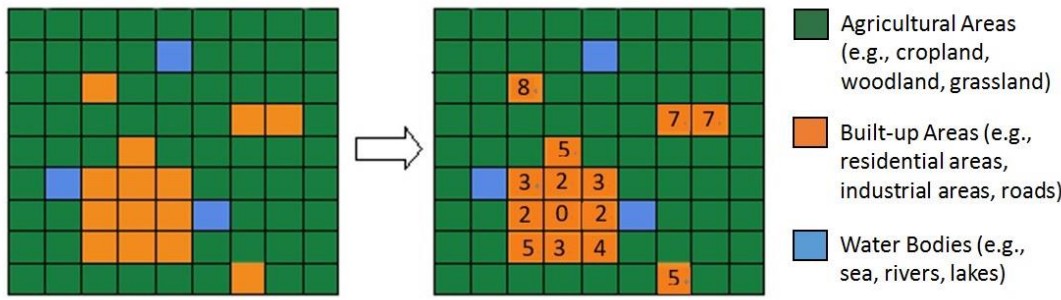

**Figure 1.** Raster land-use map and join counts values. Source: adapted from Zheng (2009).

After calculating the join counts values of all the built-up land cells, we used the ArcGIS software to match the raster land-use database with the 1:500,000 national county-level Geographic Information System (GIS) database, and then used zonal statistics tool in ArcGIS to obtain the raster land-use data of each district and county. Then, we used ArcGIS to sum up the join counts values of the built-up land cells in each district and county to obtain the total join counts values of each district and county. Following this method, we

calculated the join counts values of each district and county in 1990, 2000, 2010 and 2015. Additionally, we used the county-level land-use raster database to calculate the area of non-agricultural land in each district and county in the same period.

### 2.2.2. Landscape Metrics

Scholars often use landscape indicators to describe the ecological processes of land-use. Among them, edge indicators, including total edge and edge density, are usually used to measure the edge effect of the landscape and are considered to be closely related to the fragmentation and continuity of the landscape. A high total edge and low edge density correspond to the high continuity of the landscape [5]. Additionally, the number of patches and the landscape shape index are often used by scholars to characterize land fragmentation [22,44]. Seto and Fragkias (2005) [23] argued that the number of patches and landscape shape index can be used as measures of discrete land-use. In order to compare landscape metrics with the join counts value, following the research of some scholars [5–7,45], we selected the above four landscape metrics, that is, total edge, edge density, number of patches, and landscape shape index. Total edge means the sum of the edge lengths in the landscape involving non-agricultural land; edge density means the edge length per unit of the landscape area; number of patches means the number of patches in the landscape; and landscape shape index means the edge length weighted by the landscape area.

The FRAGSTATS software was used to calculate these landscape metrics in 1990, 2000, 2010 and 2015 for each county-level unit of China.

### 2.2.3. Other Variable

With commonly used raster land-use data, it is difficult to differentiate urban and rural land-use. This makes it challenging to quantify to what extent urban and rural expansion affected the land-use mixture. In this paper, we assume that the urbanization level (URBAN) can represent the extent of urban expansion because the land demand of the urban population is concentrated in urban areas. The urbanization level was obtained from the fifth national censuses by county in 2000. Additionally, it is believed that the regions with the fastest change of land-use in the recent years are located in the major urban agglomerations and that land-use has expanded outward from the core cities in these agglomerations [5,29]. In order to measure the impact of the distance to core cities on the land-use mixture, we calculated the distance between counties and provincial capitals (DISP). The distance was obtained from the Baidu Maps Platform API (Application Programming Interface). When more villages are present, there is often a more fragmented land-use demand; this is especially true in rural residential areas, owing to the fragmented governance in such areas [46]. Therefore, we used the number of village committees (VILLAGE) to represent the extent of rural expansion, which was obtained from the 2001 China County Statistical Yearbook.

Additionally, in order to examine the impact of institutional factors on land-use mixtures, we used two groups of institutional variables as the target independent variables, namely, land management intensity and fiscal decentralization.

As mentioned above, loose regulations encourage land-use that is not in accordance with the planning, resulting in a more serious land-use fragmentation or mixture. Following He et al. (2016) [32], we used the proportion of the number of land violation cases that were concealed and not discovered to the total number of filed land violation cases (LAW) to indicate the effort expended to investigate illegal land-uses. Additionally, the proportion of filed land violation cases to the total number of land violation cases (FILE) was used to indicate the effort to enforce land laws and regulations. For the same reason as He et al. (2016) [32], we collected data on land violation cases from 2005. The number of land violation cases were obtained from the 2006 China Land and Resource Statistical Yearbook.

As argued by He et al. (2016) [32], fiscal decentralization forces local governments to compensate fiscal gaps from land leasing, thus stimulating land urbanization, which may result in the land-use mixture. Therefore, we considered two variables, namely the fiscal gap (FGAP) and the contribution of land conveyance fees to local finance (LCF). FGAP was measured as the ratio of fiscal expenditure to fiscal revenue; the larger the fiscal gap, the more the government may emphasize land development, and thus the larger the degree of the land-use mixture will be. LCF was measured as the proportion of land transfer fees in local budget revenues, which reflects the dependence of local governments on land leasing. Reliance on land-related revenues gives local governments stronger incentives to develop land, which leads to a larger degree of land-use mixtures. The fiscal expenditures and fiscal revenues were obtained from the 2001 China County Statistical Yearbook. Land transfer fees and local budget revenues were obtained from the 2006 China Land and Resource Statistical Yearbook.

Furthermore, we added a series of control variables: (1) initial conditions, including the degree of the land-use mixture in 2000 (JC2000) and the change rate of non-agricultural land areas from 2000 to 2015 (LAND Change), which were used to control for the effects of the land-use mixture in the base period and the change in non-agricultural land on the land-use mixture. The JC2000 and LAND Change were calculated using the join counts method; (2) the globalization level, including the proportion of foreign capital to state capital (FDI) in 2000 and the proportion of the export delivery value to the industrial sales value (EXPORT) in 2000, which is closely related to land expansion. It is generally considered that areas with high levels of globalization have greater land expansion, which in turn affects the land-use mixture. These indicators were obtained from the 2000 Chinese Industrial Database; (3) socioeconomic factors, including the change rate of GDP from 2000 to 2010 (GDP Change). We assumed that the level of economic development is a basic variable that influenced land-use change, and the pursuit of economic development often leads to changes in land-use, thus affecting the land-use mixture. The GDP was obtained from the 2001 and 2011 China County Statistical Yearbook.

### 2.3. Spatiotemporal Pattern Analysis

Using the abovementioned join counts method and some general landscape metrics, we obtained the county-level data of land-use mixtures. The ArcGIS software was used to match the data with the GIS map of county-level administrative divisions to obtain the spatiotemporal pattern of the land-use mixture. The GIS map with county-level administrative divisions is based on the 1:500,000 national foundational GIS data of China for 2015.

In order to display and analyze as much as possible the trend of the spatiotemporal pattern of the land-use mixture in China over a long period of time, we drew land-use mixture maps for 1990–2000, 2000–2010, and 2010–2015, respectively. Additionally, we produced maps showing the spatiotemporal pattern of the change in a non-agricultural land area for comparative analysis.

In each map, we display two indicators, namely the absolute change and the rate of change. Among them, the absolute change is the difference between the data of two different years, and the change rate is the absolute change divided by the initial value.

Then, we conducted a detailed analysis of the spatiotemporal characteristics of the change of the non-agricultural land area and land-use mixture and summarized the characteristics of regions with the most significant changes in the above two indicators.

Additionally, the land-use mixture is usually caused by the rapid expansion of land-use. Therefore, in order to examine the change in the degree of the land-use mixture with each unit increase in the area of non-agricultural land, we calculated the ratio of the change in the join counts value to the change in the area of non-agricultural land. We also compared the above ratio of the four regions of China (Eastern, Central, Western, and Northeast China) in 1990–2000, 2000–2010, and 2010–2015 to summarize the characteristics of the land-use mixture as the land-use expands.

In order to show the changes in the land-use mixture more clearly, we additionally used landscape indicators as supplements. We drew the spatial distribution of the change in the total edge and number of patches, and calculated the ratio of the variation in the total edge and number of patches to the variation in the area of non-agricultural land in different regions of China.

### 2.4. Model Specification

In this section, we explore the driving forces of the recent change in the land-use mixture, so we selected the change rates of the land-use mixture from 2000 to 2015 as the dependent variables. This study included four dependent variables, namely the change rate of the join counts value (Join Counts Change), the edge density (Edge Density Change), the number of patches (Number of Patches Change), and the landscape shape index (Landscape Shape Index Change). The Join Count Change was calculated in two steps. The difference in the join counts value between 2000 and 2015 is calculated first and divided by the join counts value in 2000. Similar calculation methods were used for the three other dependent variables.

Following related studies [22,36], we constructed a linear regression model to examine the driving forces of land-use mixtures, which was formulated as follows:

$$\ln y_i = \beta_0 + \beta_1 \ln x_{1i} + \beta_2 \ln x_{2i} + \cdots + \beta_k \ln x_{ki} + \varepsilon_i \tag{1}$$

where $y_i$ represents the change rate of the land-use mixture in district or county $i$ from 2000 to 2015, namely the change rate of the join counts value, edge density, number of patches, and landscape shape index (i.e., Join Count Change, Edge Density Change, Number of Patches Change, and Landscape Shape Index Change); $x_{ki}$ represents the main factors affecting the change in the land-use mixture; $\beta_0$ is a constant term; $\beta_k$ is the coefficients of the independent variables; and $\varepsilon_i$ is the random error. The definitions and descriptive statistics of these variables are listed in Table 1.

**Table 1.** The definitions and descriptive statistics of the variables used in this study.

| Variable | Description | Observations | Mean | Std. Dev. |
|---|---|---|---|---|
| Join Counts Change | Change rate of join counts values from 2000–2015 | 2804 | 63.990 | 154.748 |
| Edge Density Change | Change rate of edge density from 2000–2015 | 2801 | 86.125 | 203.949 |
| Number of Patches Change | Change rate of number of patches from 2000–2015 | 2809 | 57.855 | 239.355 |
| Landscape Shape Index Change | Change rate of landscape shape index from 2000–2015 | 2809 | 19.265 | 46.383 |
| JC2000 | Degree of join counts value in 2000 | 2847 | 14,108.060 | 18,020.240 |
| LAND Change | Change rate of non-agricultural land area from 2000–2015 | 2811 | 105.588 | 226.858 |
| URBAN | Urbanization rate in 2000 | 2790 | 37.866 | 30.894 |
| VILLAGE | Number of village committees in 2000 | 2045 | 456.385 | 3542.666 |
| FGAP | Ratio of fiscal expenditures to fiscal revenue in 2000 | 2033 | 3.550 | 14.487 |
| LCF | Ratio of land conveyance fees to local budgetary revenue in 2000 | 2775 | 20.026 | 20.175 |
| LAW | Proportion of land violation cases concealed and undiscovered prior to 2005 to total number of cases filed in 2005 | 2777 | 21.910 | 18.180 |
| FILE | Proportion of land violation cases filed in cases which occurred in 2005 | 2777 | 70.568 | 19.757 |
| FDI | Ratio of foreign capital to all capital in 2000 | 2788 | 4.724 | 10.184 |
| EXPORT | Ratio of export delivery value to industrial sales value in 2000 | 2787 | 8.346 | 13.156 |
| GDP Change | Change rate of GDP from 2000–2010 | 1995 | 6924.550 | 148,860.700 |
| DISP | Distance between counties and the provincial capitals | 2838 | 278,263.300 | 273,303.000 |

Table 2 reports the correlation coefficients between the explanatory variables. It can be seen that the maximum correlation coefficient between the independent variables does not exceed 0.8. Furthermore, the variance inflation factor (VIF) of each explanatory variable was calculated, and it was found that all the VIF values were less than 6, indicating that there was no serious multicollinearity between the explanatory variables. Furthermore, to eliminate possible heteroscedasticity, the natural logarithm of each variable was taken in this study.

We first examined the impacts of the initial land-use conditions, globalization, and economic development on the land-use mixture. On this basis, this study explored the effects of urban–rural expansion and institutional factors on the land-use mixture. Furthermore, this study also analyzed the spatial heterogeneity of the land-use mixture, reporting the regression results of the effects of all explanatory variables on the land-use mixture in East China, Central China, and West China.

The methodological framework of this study is shown in Figure 2.

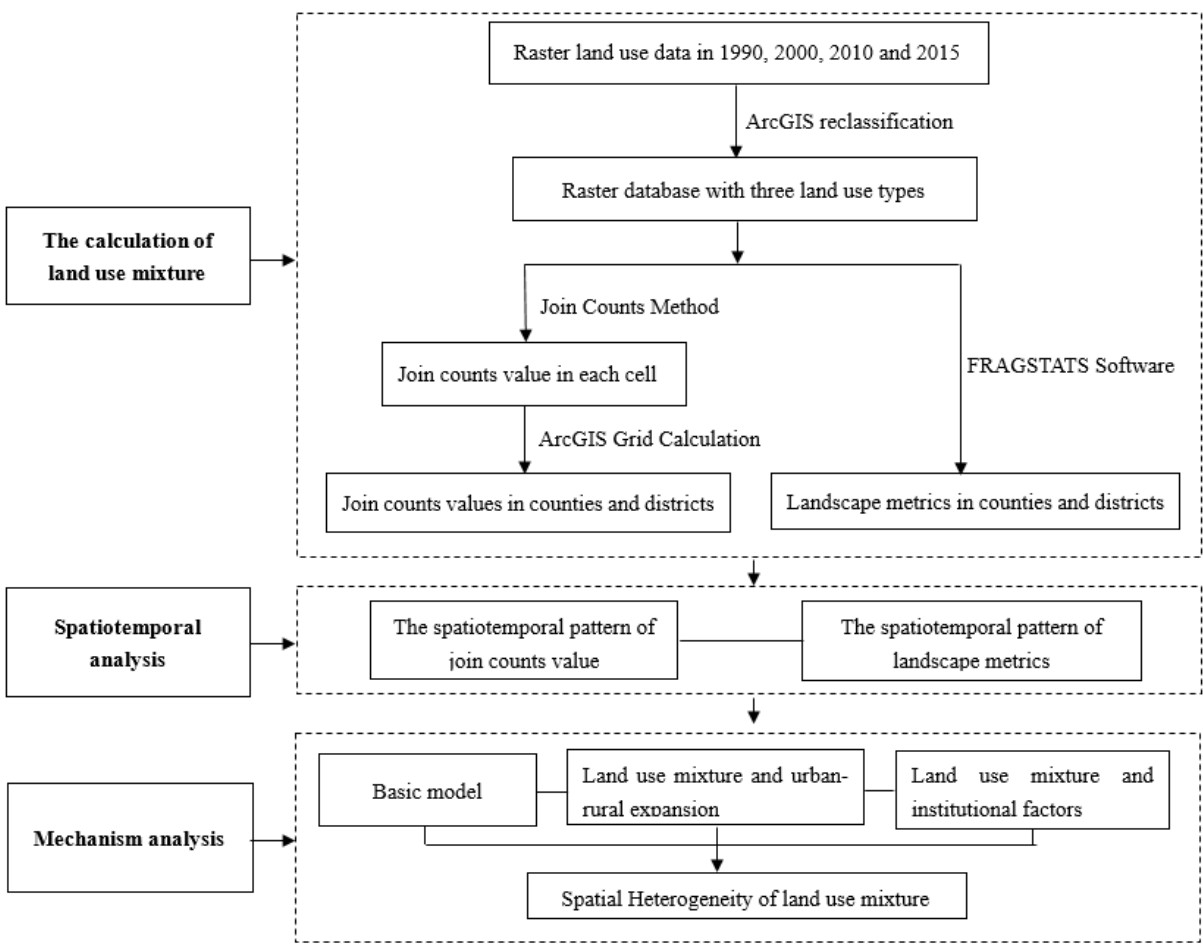

**Figure 2.** The methodological framework used in this study.

**Table 2.** The correlation matrix of the explanatory variables used in this study.

| Variable | (1) | (2) | (3) | (4) | (5) | (6) | (7) | (8) | (9) | (10) | (11) | (12) |
|---|---|---|---|---|---|---|---|---|---|---|---|---|
| (1) JC2000 | 1.000 | | | | | | | | | | | |
| (2) LAND Change | −0.059 | 1.000 | | | | | | | | | | |
| (3) URBAN | 0.067 | −0.060 | 1.000 | | | | | | | | | |
| (4) VILLAGE | 0.211 | 0.027 | −0.541 | 1.000 | | | | | | | | |
| (5) FGAP | −0.290 | 0.016 | −0.485 | 0.446 | 1.000 | | | | | | | |
| (6) LCF | 0.100 | −0.011 | 0.093 | 0.003 | −0.142 | 1.000 | | | | | | |
| (7) LAW | 0.161 | 0.185 | 0.138 | 0.004 | −0.179 | 0.030 | 1.000 | | | | | |
| (8) FILE | 0.509 | 0.048 | 0.235 | −0.016 | −0.339 | 0.073 | 0.350 | 1.000 | | | | |
| (9) FDI | 0.085 | 0.030 | 0.248 | −0.187 | −0.211 | 0.084 | 0.171 | 0.081 | 1.000 | | | |
| (10) EXPORT | 0.130 | 0.044 | 0.267 | −0.127 | −0.250 | 0.089 | 0.274 | 0.099 | 0.363 | 1.000 | | |
| (11) GDP Change | 0.079 | 0.033 | −0.525 | 0.770 | 0.548 | −0.107 | −0.096 | −0.123 | −0.233 | −0.201 | 1.000 | |
| (12) DISP | −0.006 | −0.026 | −0.122 | 0.298 | 0.335 | −0.176 | −0.053 | −0.116 | −0.163 | −0.114 | 0.352 | 1.000 |

## 3. Results

### 3.1. Spatiotemporal Pattern of Land Use Mixture

The spatiotemporal patterns of non-agricultural land and the land-use mixture (measured by the join counts method) from 1990 to 2015 at the county level are displayed in Figures 3–5.

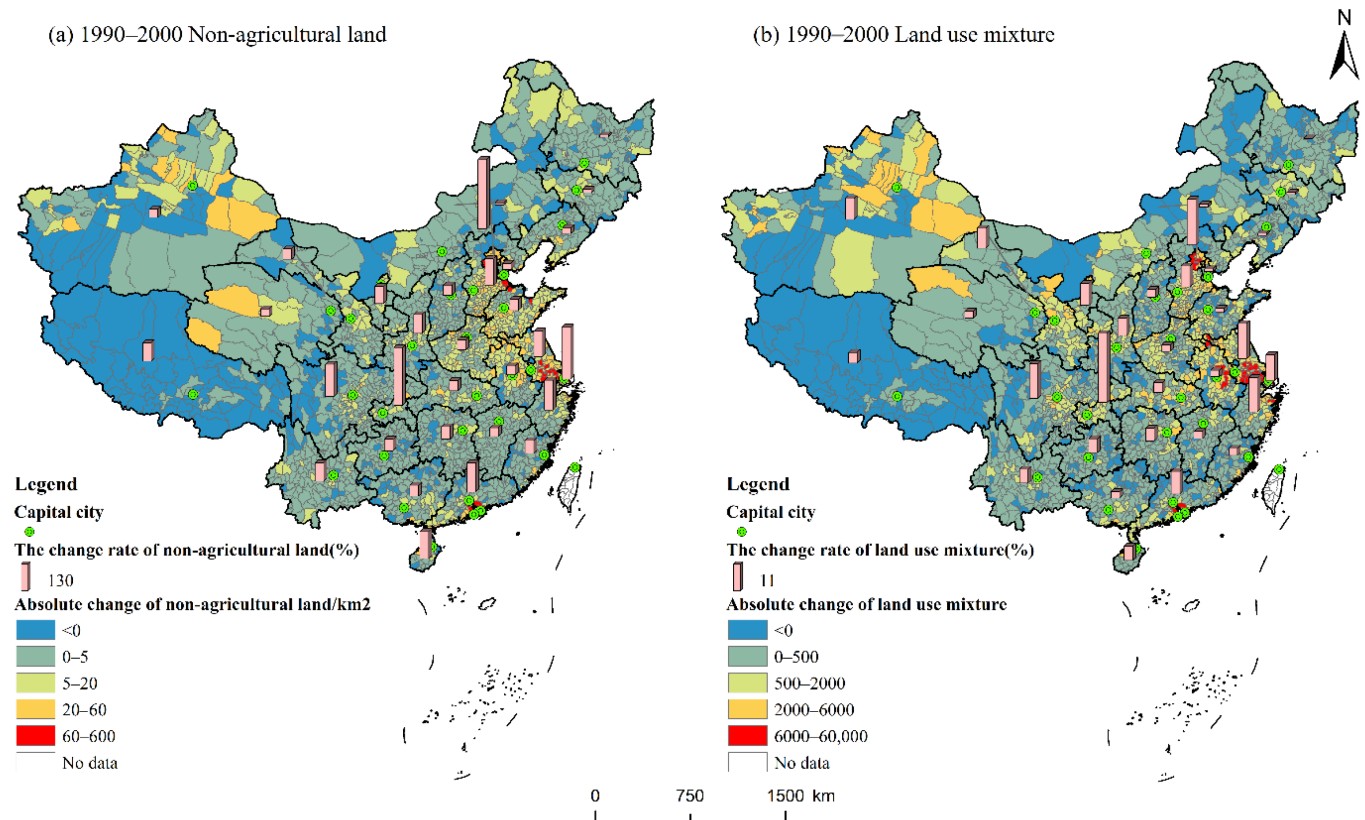

**Figure 3.** The spatiotemporal pattern of the changes in the non-agricultural land area (**a**) and land-use mixture (**b**) in 1990–2000.

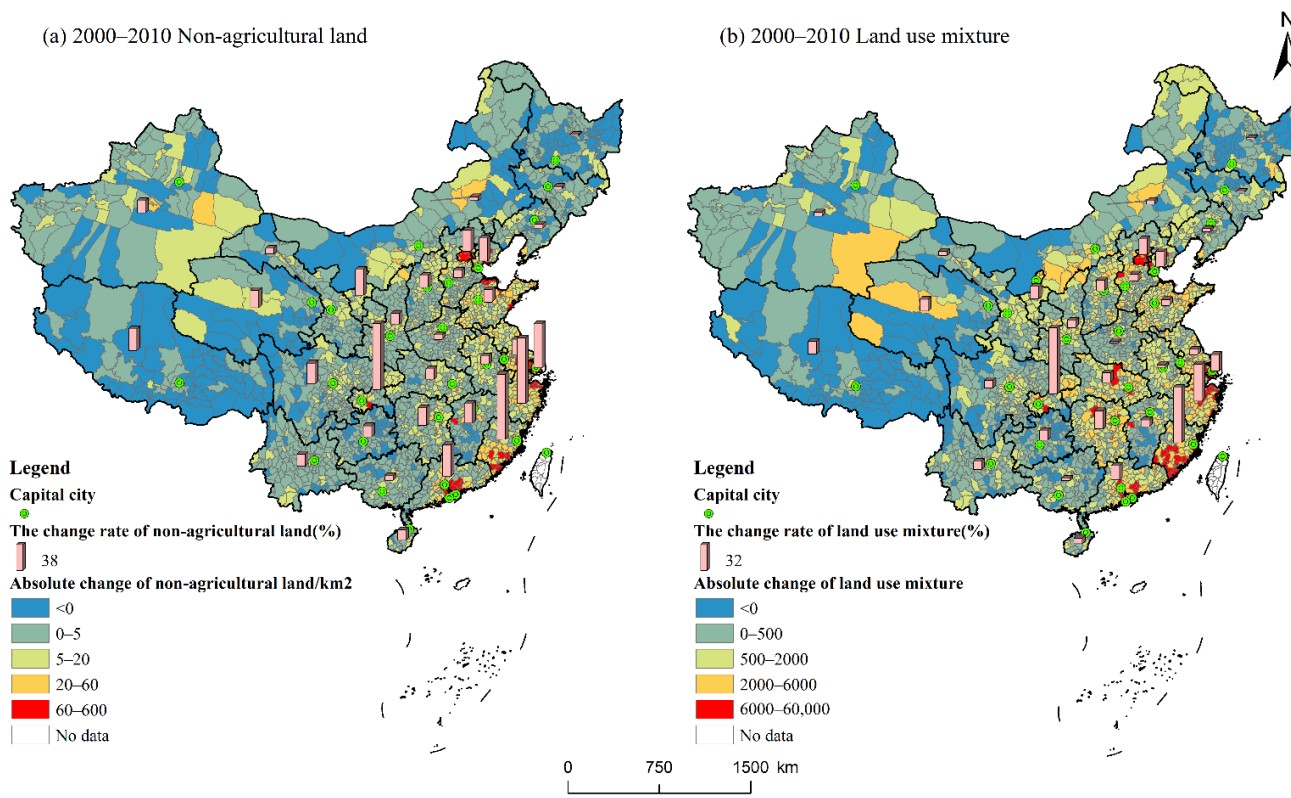

**Figure 4.** The spatiotemporal pattern of the changes in the non-agricultural land area (**a**) and land-use mixture (**b**) in 2000–2010.

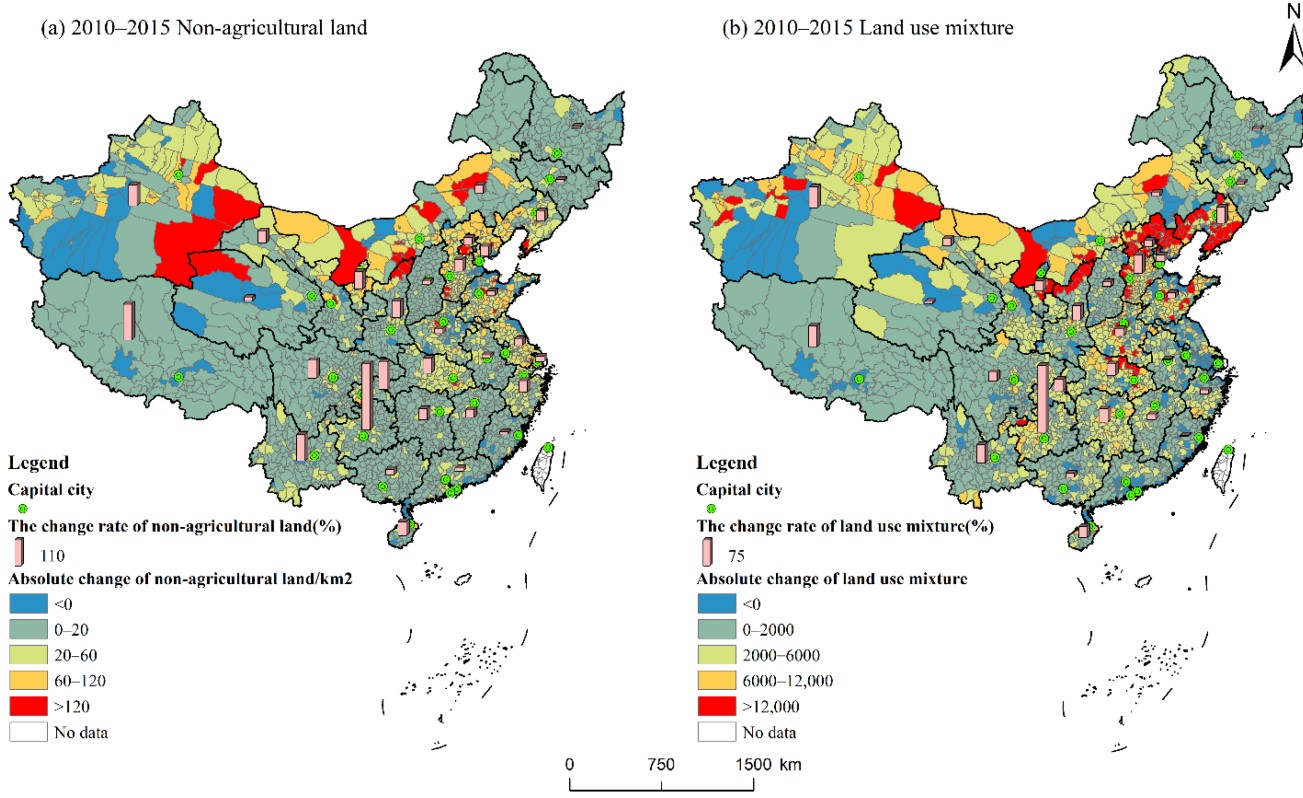

**Figure 5.** The spatiotemporal pattern of the changes in the non-agricultural land area (**a**) and land-use mixture (**b**) in 2010–2015.

The main characteristics of the spatiotemporal pattern of the change in the non-agricultural land area and land-use mixture are as follows:

The change in the non-agricultural land area during 2000–2010 was significantly higher than that during 1990–2000. During these two periods, regions with high change rates of non-agricultural land area were mainly concentrated in the Beijing-Tianjin-Hebei, Yangtze River Delta, and Pearl River Delta urban agglomerations, as well as provinces such as Chongqing and Sichuan. From 2010–2015, the non-agricultural land area in these three urban agglomerations continued to grow and the growth rate of the non-agricultural land area in Central and Western China increased significantly.

From 1990–2010, the change in trend of the land-use mixture in China was basically consistent with that of the non-agricultural land area. It should also be noted that, between 2000 and 2010, some provinces in Central China, such as Hubei and Hunan, saw prominent changes in the land-use mixture. During 2010–2015, the degree of land-use mixtures in the Yangtze River Delta and the Pearl River Delta declined, while the Beijing-Tianjin-Hebei region maintained a relatively high degree of land-use mixtures. Additionally, some provinces, such as Inner Mongolia and Liaoning, experienced a large increase in their land-use mixture during this period.

From the perspective of the spatiotemporal distribution of the land-use mixture in China at the county level, we found that the regions with significant changes in the land-use mixture have different distribution characteristics: (1) from 2000 to 2010, Zhejiang, Fujian, Chongqing and Guangdong had the highest rate of change in their land-use mixtures, with all the districts and counties in these provinces showing a high degree of land-use mixtures; from 2010–2015, Hebei, Liaoning, Inner Mongolia and Hubei, and Hunan and Henan exhibited a significant increase in land-use mixtures; (2) the districts and counties surrounding major cities experienced a relatively high rate of change in land-use mixtures, such as in the areas around Beijing; a small area around Shenyang, Anshan, and Fushun; the Shandong Peninsula coastal city belt; Jinan, Shanghai, and southern Jiangsu; the Chang-Zhu-Tan area; Wuhan; and Zhengzhou and Nanchang; and (3) non-agricultural land-use around major mining cities was also relatively mixed, such as the Yulin area in northern Shaanxi, Ordos in Inner Mongolia, Ningxia, and northern Shanxi.

By comparing the spatiotemporal distribution of the changes in non-agricultural land and the land-use mixture during 1990–2015, we found that, from 1990 to 2015, both the non-agricultural land area and land-use mixture showed an increasing trend throughout the country. However, there are some differences in the increase trends between regions: (1) from 1990 to 2010, the growth of non-agricultural land area in Hubei and Hunan was low, while the degree of the land-use mixture increased greatly; (2) from 2010 to 2015, the changes in the non-agricultural land area in Hubei, Hunan, and Henan in Central China were still small, but the degree of the land-use mixture was significantly high.

In order to examine the change in the degree of the land-use mixture with each unit increase in the area of non-agricultural land, we used the ratio of the change in the join counts value to the change in the area of the non-agricultural land (denoted as CD) (see Figure 6). From 1990–2010, the highest CD value was not observed in Eastern China but rather in Central China. Moreover, from 2010–2015, the CD values in Eastern and Western China declined significantly, while the CD values in Central and Northeast China increased significantly.

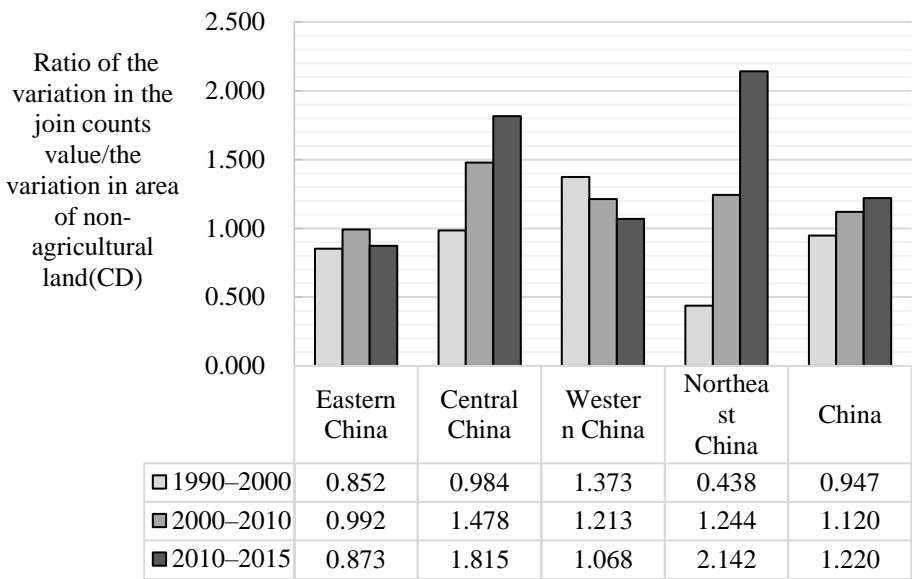

**Figure 6.** Values of the ratio of the variation in the join counts value to the variation in the area of non-agricultural land (CD) in different regions of China.

| | Eastern China | Central China | Western China | Northeast China | China |
|---|---|---|---|---|---|
| 1990–2000 | 0.852 | 0.984 | 1.373 | 0.438 | 0.947 |
| 2000–2010 | 0.992 | 1.478 | 1.213 | 1.244 | 1.120 |
| 2010–2015 | 0.873 | 1.815 | 1.068 | 2.142 | 1.220 |

We also calculated the ratio of the change in the total edge and number of patches to the change in the area of non-agricultural land from 2000–2015 for comparison (Table 3). The regions that experienced the largest increase in the total edge and number of patches per unit of non-agricultural land area are located in Central China and Northeast China, which suggests that, as the area of non-agricultural land increased in these regions, land development became more fragmented. This result is consistent with the land-use mixture shown in Figure 6. In order to further compare the spatiotemporal evolution of total edge and number of patches with the land-use mixture, we also drew maps of the changes in total edge and number of patches from 2000 to 2015 (see Figures S1 and S2 in Supplementary Materials).

**Table 3.** The ratio of the variation in total edge and number of patches to the variation in the area of non-agricultural land in different regions of China from 2000–2015.

| Region | Number of Patches | Total Edge |
|---|---|---|
| Eastern China | 0.003 | 41.454 |
| Central China | 0.015 | 73.749 |
| Western China | 0.008 | 47.797 |
| Northeast China | 0.030 | 92.438 |
| China | 0.009 | 52.014 |

### 3.2. Empirical Analysis of Land Use Mixture

#### 3.2.1. Basic Model

Table 4 presents the basic results using Join Count Change, Edge Density Change, Number of Patches Change, and Landscape Shape Index Change as dependent variables, respectively. Column (1) in Table 4 presents the regression of the Join Count Change on the initial land-use mixture (JC2000) and non-agricultural land change in 2000–2015 (LAND Change). The estimated coefficient of the degree of the join counts value in 2000 (JC2000) is significantly negative at the 1% level. This indicates that from 2000 to 2015, areas with a low degree of land-use mixtures in the early stage of development will increase more in the later stage, and vice versa. The estimated coefficient of the change rate of non-agricultural land area from 2000–2015 (LAND Change) is the largest and significantly positive at the 1% level. This result shows that a large degree of the land-use mixture is mainly caused by the transformation of land-use, which is consistent with the results of Fang et al. (2016) [5].

**Table 4.** Regression results for the impact of initial conditions on the land-use mixture.

| VARIABLE | (1) Join Count Change | (2) Join Count Change | (3) Join Count Change | (4) Edge Density Change | (5) Number of Patches Change | (6) Landscape Shape Index Change |
|---|---|---|---|---|---|---|
| JC2000 | −0.102 *** | −0.086 *** | −0.106 *** | −0.155 *** | −0.242 *** | −0.180 *** |
| | (0.011) | (0.011) | (0.012) | (0.022) | (0.020) | (0.017) |
| LAND Change | 0.820 *** | 0.827 *** | 0.817 *** | 0.652 *** | 0.621 *** | 0.445 *** |
| | (0.014) | (0.015) | (0.015) | (0.024) | (0.023) | (0.020) |
| FDI | | −0.139 *** | −0.096 *** | −0.047 ** | −0.118 *** | −0.104 *** |
| | | (0.020) | (0.019) | (0.019) | (0.027) | (0.022) |
| EXPORT | | −0.066 *** | −0.036 *** | 0.032 ** | −0.058 *** | −0.036 * |
| | | (0.014) | (0.014) | (0.014) | (0.022) | (0.021) |
| GDP Change | | | 0.108 *** | 0.004 | 0.124 *** | 0.092 *** |
| | | | (0.009) | (0.009) | (0.013) | (0.011) |
| Constant | 0.807 *** | 0.798 *** | 0.504 *** | 2.310 *** | 1.597 *** | 1.491 *** |
| | (0.126) | (0.126) | (0.145) | (0.259) | (0.214) | (0.181) |
| Observations | 2851 | 2851 | 2851 | 2851 | 2851 | 2851 |
| R-squared | 0.570 | 0.589 | 0.616 | 0.488 | 0.345 | 0.266 |

Note: Robust standard errors are given in parentheses; *** $p < 0.01$, ** $p < 0.05$, * $p < 0.1$; Join Count Change is the change rate of the join counts value from 2000–2015; Edge Density Change is the change rate of the edge density from 2000–2015; Number of Patches Change is the change rate of the number of patches from 2000–2015; Landscape Shape Index Change is the change rate of the landscape shape index from 2000–2015; JC2000 is the degree of the join counts value in 2000; LAND Change is the change rate of the non-agricultural land area from 2000–2015; FDI is the ratio of foreign capital to all capital in 2000; EXPORT is the ratio of export delivery value to industrial sales value in 2000; GDP Change is the change rate of GDP from 2000–2010.

Column (2) in Table 4 verifies the impact of globalization on land-use mixtures. FDI and the proportion of export delivery value (EXPORT) are negatively and significantly correlated with the change rate of the join counts value (Join Count Change). This result shows that those areas with a high globalization level have begun to integrate their land-use after decades of development.

The regression shown in Column (3) was performed to examine the relationship between the land-use mixture and economic development. The result shows that there is a significant positive correlation between the change rate of GDP (GDP Change) and the land-use mixture.

Using the same independent variables as Column (3), Columns (4)–(6) show the regression results using Edge Density Change, Number of Patches Change, and Landscape Shape Index Change as dependent variables, respectively. The results of Columns (4)–(6) are fairly consistent with those of Column (3), which implies that the results of the proposed model are robust. In the subsequent regressions, we used the independent variables in Table 4 as the permanent controls. The coefficients of these independent variables are not reported in the following regressions.

3.2.2. Land Use Mixture and Urban-Rural Expansion

In this section, we examine the effect of the urbanization rate (URBAN), the number of villages (VILLAGE), and the distances to provincial capitals (DISP) on the land-use mixture. Table 5 presents the regression results.

**Table 5.** Regression results for the impact of urban and rural expansion on land-use mixture.

| VARIABLE | (1) Join Count Change | (2) Join Count Change | (3) Join Count Change | (4) Edge Density Change | (5) Number of Patches Change | (6) Landscape Shape Index Change |
|---|---|---|---|---|---|---|
| URBAN | −0.153 *** | −0.104 *** | −0.122 *** | 0.047 * | −0.092 ** | −0.047 |
| | (0.023) | (0.022) | (0.025) | (0.026) | (0.040) | (0.036) |
| VILLAGE | | 0.125 *** | 0.122 *** | 0.068 *** | 0.126 *** | 0.105 *** |
| | | (0.018) | (0.018) | (0.017) | (0.028) | (0.026) |
| DISP | | | 0.109 *** | 0.032 | 0.152 *** | 0.148 *** |
| | | | (0.020) | (0.024) | (0.026) | (0.025) |
| Constant | 1.045 *** | 1.017 *** | −0.187 | 1.845 *** | 0.295 | 0.054 |
| | (0.167) | (0.164) | (0.285) | (0.416) | (0.385) | (0.367) |
| Observations | 2851 | 2851 | 2851 | 2851 | 2851 | 2851 |
| R−squared | 0.621 | 0.628 | 0.634 | 0.491 | 0.362 | 0.286 |

Note: All of the variables were controlled in Table 4, and their results were not reported in this table to save space. Robust standard errors are given in parentheses; *** $p < 0.01$, ** $p < 0.05$, * $p < 0.1$; Join Count Change is the change rate of the join counts value from 2000–2015; Edge Density Change is the change rate of the edge density from 2000–2015; Number of Patches Change is the change rate of the number of patches from 2000–2015; Landscape Shape Index Change is the change rate of the landscape shape index from 2000–2015; URBAN is the urbanization rate in 2000; VILLAGE is the number of village committees in 2000; DISP is the distance between counties and the provincial capitals.

As shown in the first column, the coefficient of the urbanization rate (URBAN) is significantly negative. This regression suggests that the urbanization rate is negatively correlated with the land-use mixture. Meanwhile, as shown in the second column, the coefficient of the number of villages (VILLAGE) is significantly positive, that is, the more villages, the more mixed the land-use, which is consistent with the findings of Lin (2007) [31]. Moreover, as shown in the third column, there is a significant negative influence of the distance to provincial capital cities (DISP) on the Join Counts Change, indicating that the farther away from the capital city, the more mixed the land-use.

Columns (4)–(6) show the regression results using Edge Density Change, Number of Patches Change, and Landscape Shape Index Change as dependent variables, respectively. The results of the regressions in Columns (4)–(6) are fairly consistent with those of the regression in Column (3).

### 3.2.3. Land-Use Mixture and Institutional Factors

Table 6 presents the regression results after adding institutional factors. The first column presents the regression of the effect of fiscal decentralization on Join Counts Change. The estimated coefficients of the fiscal gap (FGAP) and the contribution of land conveyance fees to local finance (LCF) are significantly negative at the 1% level, indicating that the increase in the fiscal gap and land conveyance may lead to a decline in the land-use mixture. The second column presents the regression of the effect of land management intensity on Join Count Change. The coefficient of the proportion of concealed land violation cases (LAW) is significantly negative, and the coefficient of the proportion of filed land violation cases (FILE) is significantly positive, which shows that the enforcing of land laws and regulations can reduce the degree of the land-use mixture. The regression in Column (3) adds all of the four institutional variables, and the result is consistent with the regressions in the first two columns. Additionally, we also used Edge Density Change, Number of Patches Change, and Landscape Shape Index Change as dependent variables, respectively, as shown in the regressions in Columns (4)–(6). The results are consistent with the regression in Column (3), which proves the robustness of the regression results.

**Table 6.** Regression results for the impact of institutional factors on the land-use mixture.

| VARIABLE | (1)<br>Join Count<br>Change | (2)<br>Join Count<br>Change | (3)<br>Join Count<br>Change | (4)<br>Edge Density<br>Change | (5)<br>Number of<br>Patches Change | (6)<br>Landscape Shape<br>Index Change |
|---|---|---|---|---|---|---|
| FGAP | −0.125 *** | | −0.089 ** | −0.100 ** | 0.034 | −0.049 |
| | (0.038) | | (0.035) | (0.044) | (0.061) | (0.053) |
| LCF | −0.067 *** | | −0.064 *** | −0.040 ** | −0.095 *** | −0.049 ** |
| | (0.018) | | (0.018) | (0.018) | (0.029) | (0.025) |
| LAW | | −0.049 *** | −0.053 *** | −0.030 * | 0.104 *** | 0.067 *** |
| | | (0.018) | (0.018) | (0.017) | (0.025) | (0.025) |
| FILE | | 0.324 *** | 0.316 *** | 0.454 *** | 0.310 *** | 0.349 *** |
| | | (0.035) | (0.035) | (0.047) | (0.054) | (0.045) |
| Constant | 0.140 | −0.832 *** | −0.526 * | 1.125 *** | −0.164 | −0.538 |
| | (0.295) | (0.271) | (0.281) | (0.397) | (0.408) | (0.377) |
| Observations | 2851 | 2851 | 2851 | 2851 | 2851 | 2851 |
| R−squared | 0.637 | 0.647 | 0.649 | 0.528 | 0.381 | 0.311 |

Note: All of the variables were controlled in Table 5, and their results were not reported in this table to save space. Robust standard errors are given in parentheses; *** $p < 0.01$, ** $p < 0.05$, * $p < 0.1$; Join Count Change is the change rate of the join counts value from 2000–2015; Edge Density Change is the change rate of the edge density from 2000–2015; Number of Patches Change is the change rate of the number of patches from 2000–2015; Landscape Shape Index Change is the change rate of the landscape shape index from 2000–2015; FGAP is the ratio of fiscal expenditures to fiscal revenue in 2000; LCF is the ratio of land conveyance fees to local budgetary revenue in 2000; LAW is the proportion of land violation cases concealed and undiscovered prior to 2005 to the total number of cases filed in 2005; FILE is the proportion of land violation cases filed in cases that occurred in 2005.

### 3.3. Spatial Heterogeneity of Land Use Mixture

In this section, we examine the regional differences in the impact of urban and rural expansion and institutional factors on the land-use mixture. Table 7 presents the regression results for Eastern, Central, and Western China.

The regressions in Columns (1), (5), and (9) present the results of the effect of targeted variables on the change rate of the join counts value (Join Count Change) in Eastern, Central, and Western China, respectively. The coefficients of the urbanization rate (URBAN), number of villages (VILLAGE), distance to provincial capitals (DISP), fiscal gap (FGAP), and land conveyance fees (LCF) in different regions are consistent with the result for the whole of China as shown in Tables 5 and 6. This suggests that, regardless of whether the region is developed or underdeveloped, the urbanization rate is significantly negatively correlated with the land-use mixture and rural expansion is significantly positively correlated with the land-use mixture. Moreover, regardless of whether the region is developed or underdeveloped, the increase in the fiscal gap and land conveyance fees does not always lead to an increase in the degree of the land-use mixture.

However, according to the coefficients of the proportion of concealed land violation cases (LAW) and the proportion of filed land violation cases (FILE) in Columns (1), (5), and (9), institutional factors have different effects on the land-use mixture in different regions. For example, in Eastern China, the estimated coefficient of LAW is significantly negative, and the coefficient of FILE is not significant, which indicates that the effort expended to investigate illegal land-use could have a significant impact on the land-use mixture, that is, strict land control tends to decrease the land-use mixture. In Central and Western China, the coefficients of LAW and FILE are significantly positive. This indicates that the effort expended to investigate illegal land-use and enforce land laws and regulations cannot yet decrease the degree of the land-use mixture in these less developed regions, that is, land-use management does not play a significant role in reducing the land-use mixture in these regions.

Additionally, we also present the results for different regions using Edge Density Change, Landscape Shape Index Change, and Number of Patches Change as dependent variables, respectively (see Table 7). The results are consistent with the regression results in Columns (1), (5), and (9). This suggests that land management plays an important role in the process of the land-use mixture, and that its influence varies between regions.

**Table 7.** Regression results for the land-use mixture in Eastern, Central, and Western China.

| VARIABLE | Eastern China | | | | Central China | | | | Western China | | | |
|---|---|---|---|---|---|---|---|---|---|---|---|---|
| | (1) | (2) | (3) | (4) | (5) | (6) | (7) | (8) | (9) | (10) | (11) | (12) |
| | Join Count Change | Edge Density Change | Number of Patches Change | Landscape Shape Index Change | Join Count Change | Edge Density Change | Number of Patches Change | Landscape Shape Index Change | Join Count Change | Edge Density Change | Number of Patches Change | Landscape Shape Index Change |
| URBAN | −0.475 *** | −0.242 *** | −0.411 *** | −0.513 *** | −0.294 *** | −0.017 | −0.299 *** | −0.091 | −0.078 ** | −0.009 | 0.038 | −0.036 |
| | (0.076) | (0.073) | (0.096) | (0.091) | (0.061) | (0.055) | (0.103) | (0.102) | (0.033) | (0.036) | (0.058) | (0.050) |
| VILLAGE | 0.158 *** | 0.051 | 0.051 | 0.027 | 0.125 *** | 0.053 | 0.285 *** | 0.200 *** | 0.074 *** | 0.042 * | 0.081 ** | 0.056 |
| | (0.039) | (0.036) | (0.059) | (0.052) | (0.034) | (0.037) | (0.073) | (0.066) | (0.023) | (0.022) | (0.041) | (0.036) |
| DISP | 0.155 *** | 0.121 *** | 0.208 *** | 0.197 *** | 0.190 *** | 0.046 | 0.070 | 0.172 *** | 0.147 *** | 0.094 *** | 0.215 *** | 0.221 *** |
| | (0.040) | (0.044) | (0.046) | (0.042) | (0.055) | (0.044) | (0.071) | (0.064) | (0.032) | (0.036) | (0.047) | (0.041) |
| FGAP | −0.057 | −0.167 | 0.677 *** | 0.198 | −0.204 ** | −0.147 | −0.392 ** | −0.226 | −0.048 | −0.041 | 0.028 | −0.005 |
| | (0.108) | (0.112) | (0.155) | (0.144) | (0.095) | (0.123) | (0.177) | (0.152) | (0.040) | (0.047) | (0.069) | (0.059) |
| LCF | −0.174 *** | −0.171 *** | −0.218 *** | −0.155 ** | −0.119 * | −0.046 | −0.145 | −0.087 | −0.070 *** | −0.052 ** | −0.116 *** | −0.085 *** |
| | (0.062) | (0.058) | (0.079) | (0.072) | (0.064) | (0.067) | (0.099) | (0.092) | (0.019) | (0.021) | (0.035) | (0.030) |
| LAW | −0.356 *** | −0.336 *** | −0.440 *** | −0.395 *** | 0.099 *** | 0.043 | 0.162 *** | 0.258 *** | 0.000 | 0.018 | 0.327 *** | 0.156 *** |
| | (0.062) | (0.070) | (0.079) | (0.076) | (0.025) | (0.027) | (0.043) | (0.043) | (0.026) | (0.037) | (0.049) | (0.043) |
| FILE | −0.076 | 0.224 | −0.587 *** | −0.462 *** | 0.785 *** | 1.115 *** | 2.246 *** | 2.002 *** | 0.161 *** | 0.245 *** | −0.042 | 0.077 |
| | (0.130) | (0.142) | (0.197) | (0.159) | (0.196) | (0.262) | (0.445) | (0.376) | (0.046) | (0.074) | (0.071) | (0.056) |
| Constant | 4.098 *** | 5.174 *** | 7.457 *** | 7.284 *** | −3.727 *** | −2.478 * | −6.996 *** | −8.859 *** | −1.473 *** | −0.263 | −2.220 *** | −2.082 *** |
| | (0.824) | (1.035) | (1.202) | (1.061) | (1.221) | (1.428) | (2.346) | (2.040) | (0.459) | (0.616) | (0.681) | (0.602) |
| Observations | 770 | 770 | 770 | 770 | 708 | 708 | 708 | 708 | 1,086 | 1,086 | 1,086 | 1,086 |
| R-squared | 0.542 | 0.352 | 0.356 | 0.340 | 0.667 | 0.475 | 0.327 | 0.373 | 0.788 | 0.723 | 0.541 | 0.481 |

Note: All of the variables were controlled in Table 4, and their results were not reported in this table to save space. Robust standard errors are given in parentheses; *** $p < 0.01$, ** $p < 0.05$, * $p < 0.1$; Join Count Change is the change rate of the join counts value from 2000–2015; Edge Density Change is the change rate of the edge density from 2000–2015; Number of Patches Change is the change rate of the number of patches from 2000–2015; Landscape Shape Index Change is the change rate of the landscape shape index from 2000–2015; URBAN is the urbanization rate in 2000; VILLAGE is the number of village committees in 2000; DISP is the distance between counties and the provincial capitals; FGAP is the ratio of fiscal expenditures to fiscal revenue in 2000; LCF is the ratio of land conveyance fees to local budgetary revenue in 2000; LAW is the proportion of land violation cases concealed and undiscovered prior to 2005 to the total number of cases filed in 2005; FILE is the proportion of land violation cases filed in cases that occurred in 2005.

## 4. Discussion

### 4.1. Spatial Characteristics of Land-Use Mixture

In Section 3.1, it was shown that, in China, the degree of the land-use mixture appears to have increased between 1990 and 2015. This phenomenon is consistent with widely occurring rural–urban hybrid in Global South contexts, such as Indonesia and Malaysia [8,47]. Meanwhile, we found that our results were inconsistent with the findings of Fang et al. (2016), who found an overall decrease in the fragmentation of the urban landscape in China between 1990 and 2005. This may be attributed to, (1) when the number and average area of patches increased as Fang et al. (2016) [5] calculated, the degree of fragmentation calculated based on patches may have decreased if urban areas expanded and merged into a continuous urban fabric; (2) considering the land-use mixture in our study, the invasion of urban land into rural areas may have led to an increase in the degree of the mixture.

Additionally, Figures 3–5 show the following characteristics: (1) from 2000–2010, Zhejiang and Fujian provinces underwent a rapid change in the transformation and mixture of land-use. This may be the result of the active development of the private economy. Although Chongqing has experienced rapid economic development and non-agricultural transformation, Chongqing's topography has made it difficult to intensively use its non-agricultural land, which in turn has led to the land-use mixture. In Guangdong, since 2000, the provincial government has advocated industrial transfer from the Pearl River Delta to the north. As a result, land-use in northern Guangdong has increased greatly, and the land-use mixture has increased in the mountainous areas [48]; (2) from 2010–2015, Henan, Hubei and Hunan in the Central region exhibited a significant increase in the land-use mixture, which was due to the industrial transfer from the Eastern region. Since 2010, the Eastern region transferred a large number of industries to the Central regions in a dispersed manner [49], which resulted in the scattered land-use. Moreover, in order to undertake the transfer of industries in the Eastern region, a large amount of land development and expansion has been carried out in the Central region, which indirectly led to mixed land-use. Apart from central provinces, Inner Mongolia, Hebei and Liaoning also experienced fast land-use mixtures. According to some scholars, from 2010–2015, due to the development of urbanization, as well as the development of mineral resources, the construction land in these provinces increased significantly [50], which led to the mixture of land-use; (3) some major cities witnessed the fastest growth in the urban economy in China since 2000. Both real estate development and the expansion of industrial zones tended to occur predominantly in the peripheries of major cities [51]. As a result, the districts and counties surrounding major cities had a relatively high degree of land-use mixtures; (4) driven by heavy industrial development, the mineral-rich regions of China developed rapidly after 2000; these include Inner Mongolia, whose economy grew at an average annual rate of 23% from 2000 to 2010. During this period, a large amount of agricultural land was converted into non-agricultural land. The dispersed spatial characteristics of mining development and extensive local real estate development that occurred may explain the highly mixed use of non-agricultural land in these regions; and (5) according to the CD values calculated in the paper, the degree of the land-use mixture in Eastern and Western China declined, while that in Central and Northeast China increased significantly. The calculation results using other landscape indicators were consistent with the results of the CD values, indicating that the join counts method we used is reliable.

### 4.2. Factors Affecting Land Use Mixture

Our results demonstrate that the land-use mixture is closely related to the increase in non-agricultural land area. In other words, the increase in construction land leads to an increase in the land-use mixture, which is consistent with the reality of the land-use change in China. Additionally, the degree of the join counts value in 2000 is negatively correlated with the land-use mixture. In 2000, places with a high degree of land-use mixtures were usually areas with rapid rural industrialization, especially those located in the Yangtze River Delta and the Pearl River Delta. After 2000, with the improvement of government

awareness of land planning and management, measures for controlling land-use have gradually been taken, thus reducing the degree of land-use mixtures. On the contrary, the level of land-use mixture in Central China was relatively low in 2000. Subsequently, along with industrial transfer from Eastern China to Central China, Central China experienced rapid industrialization and active land transformation, and, in the absence of planning and management experience, the degree of the land-use mixture has increased significantly. These results suggest that the land-use mixture in Central China changed considerably from 2010 to 2015.

Moreover, our results indicate that globalization has reduced the degree of land-use mixtures in China. The Yangtze River Delta and Pearl River Delta regions, which have the most prominent export-oriented economies in China, have begun to integrate their land-use after decades of development, resulting in a decline in the degree of land-use mixtures. Economic growth is usually accompanied by an increase in land-use, thereby intensifying the land-use mixture, which is consistent with previous studies [11,40].

By examining the impacts of urban and rural expansion on the land-use mixture, we found that there are differences in the roles of urban and rural expansion. A high level of urbanization is negatively correlated with the land-use mixture. Areas with high levels of urbanization tend to have high levels of management and strict planning, which could reduce the degree of the land-use mixture. Additionally, the positive effect of the distance between counties and provincial capitals on the land-use mixture also reflects the role of planning and management in the mixture. The closer a county is to the provincial capital city, the stricter the land-use planning and management will be, and the lower the degree of mixed land-use will be. In contrast, counties far away from provincial capitals have more mixed land-use, which can be attributed to their being beyond the reach of provincial governments' planning and management.

In rural areas, villages are scattered throughout the countryside and are managed by different village committees. When these scattered villages expand outward, in the absence of unified planning, they often convert a large amount of farmland into construction land, resulting in mixed land-use. The more village committees there are, the more agricultural land will be converted to construction land, and the more mixed the land-use will be. In recent years, village relocation and combination have been implemented all over China, and it is believed that these practices can play a positive role in reducing the waste of land resources in the future.

The results also indicate that the fiscal gap and land conveyance fees are significantly negatively correlated with land-use mixtures. According to the fiscal gap data used in this paper, regions with large fiscal gaps were mainly located in underdeveloped regions in China. These underdeveloped regions have less land development, which may result in a low degree of mixed land-use. Additionally, the data on land conveyance fees in China suggest that places with high fees tend to be economically developed tourist areas, such as Sichuan and Hainan. These areas tend to have better development planning and management, so the increase in land conveyance fees did not increase the land-use mixture. The results of this study suggest that the expansion of the fiscal gap and additional land conversion do not necessarily lead to the mixture of land-use. These conclusions are inconsistent with those of He et al. (2016) [32], who concluded that land conveyance fees and the fiscal gap index had a positive impact on the change rate of urban land-use. Nonetheless, we believe that our findings are reasonable. The present findings indicate that the demand for land finance does lead to an increase in land conversion; however, they also suggest that the growth of land-use need not be in a mixed form.

Furthermore, our results indicate that land planning and management play a vital role in mixed land-use, and additionally suggest that the impact of land management on the land-use mixture differs significantly between different regions in China. In Eastern China, which is relatively developed, the investigation and punishment of illegal land-use have played a role in reducing mixed land-use. This result is consistent with the findings of Tian (2015) [7]. Since the beginning of China's reform and opening-up policies, Eastern China

has accumulated a large amount of experience in land planning and management. With the rapid economic growth, non-agricultural land quotas in Eastern China (especially in large cities) have been strictly controlled. Meanwhile, the central government has increased the intensity of the cleaning-up and reorganizing of development zones (kaifa qu in Mandarin Chinese) in this region [52,53]. Under these circumstances, strict land-use management has decreased the mixture of land-use in Eastern China. However, since Central and Western China are in the early stages of industrialization and urbanization, land planning and management struggle to restrain land-use mixtures caused by rapid development and active land transformation. To support industrial development, the central government has given large land quotas to Central and Western China, ignoring management guidelines on the establishment of development zones, which has resulted in a significant deterioration of land-use efficiency in these regions [52,53]. Therefore, the lack of land management is the root cause of the land-use mixture.

To summarize, our study showed that the distribution of the land-use mixture in China is similar to the land-use distribution found in previous studies. However, the results of this work indicate that there are differences between the mechanisms of the land-use mixture and land-use expansion. Previous studies often attributed the expansion of non-agricultural land to rapid industrialization and urbanization [28,29]. However, our results indicate that urbanization does not necessarily lead to a land-use mixture, especially urbanization with formal land-use planning and management. Additionally, many studies attributed China's land-use expansion to land financing by local governments [32]. However, our study found that it was the management level of local governments, not land financing, that led to the land-use mixture.

## 5. Conclusions

According to the related literatures, the urban–rural juxtaposition revealed by the desakota model occurs mainly in the densely populated areas [14,16]. Our study found that the urban–rural land-use mixture demonstrated in the desakota model is present almost throughout China. In the current stage of China's industrialization under the condition of telecommunication technology, industrialization does not necessarily occur in cities, but can occur at a large scale in rural areas. This is applicable all over China. So, a large number of industrial plants have appeared in rural areas, and a lot of agricultural lands are invaded by construction land, resulting in a land-use mixture.

This paper demonstrates the spatial–temporal evolution of the land-use mixture and its mechanisms. This phenomenon of mixed land-use is not only found in major urban agglomerations, but also in the newly developed central region. In the early stage of development, some regions may pay excessive attention to economic development and neglect the reasonable planning of land-use, leading to the mixture of land-use and heavy costs. It is suggested that land planning and management should be performed in the early stage of land development to promote sustainable development.

Some researchers may argue that gathering plants together will increase the production costs of these plants and hinder the willingness of manufacturers to build plants in the region, thus negatively affecting the local economy. However, a region may be more attractive to plants if the local government can provide proactive services to help solve the problems associated with the entry and operation of plants and create a good business environment. As suggested by research using the Land Sparing-Sharing Model [54], policy may induce some companies and individuals to actively develop land, which may hurt the interests of others, such as farmers. Therefore, the spatial optimization of land-use should be prudently operated and planned by governments to ensure the rights of every stakeholder.

This paper is not intended to give a definite conclusion for the current situation of mixed land-use in China, but rather to present a quantitative examination of the land-use mixture in China so as to provide a possible basis for regional policymaking.

Due to limited data availability, we did not separately classify non-agricultural land-use in this study, which hindered our ability to accurately depict the mixture of urban

and rural land-use, especially for industrial land-use. Based on the available data, our calculations of mixed land-use did not consider the influence of scale, which may affect the results. In the future, with further improvements in remote sensing image acquisition and interpretation for land-use, the mixture of industrial land and residential land may be presented more accurately, and multi-scale analyses may be improved.

**Supplementary Materials:** The following are available online at https://www.mdpi.com/article/10.3390/land10040370/s1, Text S1: The contents of 25 types of land use, Figure S1: The spatiotemporal pattern of the changes in total edge in 2000–2015, Figure S2: The spatiotemporal pattern of the changes in number of patches in 2000–2015.

**Author Contributions:** Conceptualization, Y.Z.; Methodology, Y.Z.; Data Curation, S.Z.; Software, A.L.; Writing—Original draft preparation, S.Z.; Writing—Reviewing and Editing, J.H.; Supervision, Y.Z. and A.L. All authors have read and agreed to the published version of the manuscript.

**Funding:** This research was supported by the National Key Research and Development Program of China (2017YFC1503002), the National Natural Science Foundation of China (41001094), the Beijing Key Lab of Study on SCI-TECH Strategy for Urban Green Development, the Important Science & Technology Specific Projects of Qinghai Province (2019-SF-A4-1) and the National Natural Science Foundation of Qinghai Province (2019-ZJ-7020).

**Data Availability Statement:** The data presented in this study are available on request from the corresponding author.

**Conflicts of Interest:** The authors declare that we have no known competing financial interest or personal relationships that could have influenced the work reported in this paper.

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
