# Peer review of "Analysis of the Spatiotemporal Pattern and Mechanism of Land Use Mixture: Evidence from China’s County Data"

_land, doi:10.3390/land10040370_

Round 1

Reviewer 1 Report

The paper analyses the land use mixtures and their accompanying mechanisms in China. This topic is a very relevant topic especially for developing countries, where this problem is very pronounced. However, in the paper, the authors focus their work on China. And though a discussion is undertaken to place the work in the wider scope of literature, the literature discussed in the section, and in fact in all of the manuscript are from China. This severely limits the study which will very much benefit from a discussion of the results from the stand point of other countries or regions in the world, with a similar problem, but a different social & cultural context.

The manuscript should also be proof read to correct the english grammer and spellings.

Author Response

Point 1: The paper analyses the land use mixtures and their accompanying mechanisms in China. This topic is a very relevant topic especially for developing countries, where this problem is very pronounced. However, in the paper, the authors focus their work on China. And though a discussion is undertaken to place the work in the wider scope of literature, the literature discussed in the section, and in fact in all of the manuscript are from China. This severely limits the study which will very much benefit from a discussion of the results from the stand point of other countries or regions in the world, with a similar problem, but a different social & cultural context.

Response 1: Thank you for your suggestion. In the revised manuscript, we have added some comparison between the results of our study and foreign related studies on land use mixture in the discussion section and conclusions section in lines 632–636 and lines 806–813. We also reviewed some foreign literatures in the introduction section in lines 71–75.

Point 2: The manuscript should also be proof read to correct the English grammar and spellings.

Response 2: Thank you very much for your advice. In the revised manuscript, we have checked and revised the English grammar and spellings carefully. Additionally, our paper has been edited by a native-English professional science editor and major language revisions have been made.

Reviewer 2 Report

Comments can be found in the PDF document

Reviewer 3 Report

Dear Authors,

Several aspects should be edited and amplified.

In this regard, comments according to the different parts of the text are going to be described.

  1. The abstract should be more concrete, sharp, and critical.

  1. The introduction does not provide enough theoretical support for the subject.

  1. Highlight the aspects in which your research departs from the existing literature.

  1. What are the innovative contributions of your manuscript to science?

  1. You should include a case study contextualization. All elements that could help the reader better picture what looks like the area and the phenomena you are talking about (e.g. GDP, population density, land use structure, etc). I believe one of the aims of the work is to make readers understand the magnitude of the issue authors are debating.

  1. Please include a methodological framework.

  1. Table 1. The use of each variable should be justified.

  1. Why did you use total edge (TE), edge density (ED), number of patches (NP) and landscape shape index as landscape metrics, instead of other metrics? (such as connectivity or isolation/ proximity). This needs better justification.

  1. The maps in figures 2, 3, 4, 5, 6 and 7 should be better organized.

  1. The discussion section needs substantial revise to include an in-depth discussion on those important results including compare your results with previous studies and explain better why your results are similar or differences from previous findings.

  1. Add in conclusion more own opinion, critical thinking and what is the limitation of the study? In conclusion, you can address the flaws and limitations of your approach.

Round 2

Reviewer 2 Report

the revised manuscript has been considerably improved

Reviewer 3 Report

Thank you and your colleagues for the modifications that you have made to this article and how well you have responded to the suggestions.